# Diverse Functions of Multiple *Bdnf* Transcripts Driven by Distinct *Bdnf* Promoters

**DOI:** 10.3390/biom13040655

**Published:** 2023-04-06

**Authors:** He You, Bai Lu

**Affiliations:** 1School of Pharmaceutical Sciences, IDG/McGovern Institute for Brain Research, Tsinghua University, Beijing 100084, China; youh13@tsinghua.org.cn; 2School of Life Sciences, Tsinghua University, Beijing 100084, China; 3Stellenbosch Institute for Advanced Study (STIAS), Wallenberg Centre, 10 Marais Street, Stellenbosch 7600, South Africa

**Keywords:** brain-derived neurotrophic factor, non-coding exons, *Bdnf* promoters

## Abstract

The gene encoding brain-derived neurotrophic factor (*Bdnf*) consists of nine non-coding exons driven by unique promoters, leading to the expression of nine *Bdnf* transcripts that play different roles in various brain regions and physiological stages. In this manuscript, we present a comprehensive overview of the molecular regulation and structural characteristics of the multiple *Bdnf* promoters, along with a summary of the current knowledge on the cellular and physiological functions of the distinct *Bdnf* transcripts produced by these promoters. Specifically, we summarized the role of *Bdnf* transcripts in psychiatric disorders, including schizophrenia and anxiety, as well as the cognitive functions associated with specific *Bdnf* promoters. Moreover, we examine the involvement of different *Bdnf* promoters in various aspects of metabolism. Finally, we propose future research directions that will enhance our understanding of the complex functions of *Bdnf* and its diverse promoters.

## 1. The Unique Genomic Structure of the *Bdnf* Gene and Its Transcripts

### 1.1. The Genomic Structure of Bdnf

*Bdnf* is a eukaryotic gene with a unique genomic structure that is very different from the majority of other genes. It consists of multiple 5′ non-coding exons, also known as 5′untranslated regions (5′-UTRs), and one 3′ exon that codes for the pre-pro-BDNF protein. In rodents, nine functional promoters have been identified upstream of the nine non-coding exons, numbered I to IX or e1 to e9. Promoters I to VIII initiate transcription from the 5′ non-coding Exons I to VIII, which are spliced onto the common coding exon (Exon IX) to generate eight mRNA transcripts. Promoter IX, on the other hand, initiates transcription directly from the 5′-UTR region of the coding Exon IX (Figure 1A).

In humans, there are also nine functional promoters, but there are 11 exons, including 10 non-coding exons and one 3′ coding exon (Figure 1B). DNA sequences of the human *BDNF* exons homologous to those of rodent exons are named with the same number from I to IX, while exons unique to *BDNF* are labeled with the number of neighboring exons and the letter ‘h’ as a suffix. For example, human Exon Vh represents the unique human exon neighboring Exon V. Among the 11 exons, Exons VIII and VIIIh are not linked to separate functional promoters. These two exons can be alternatively spliced with Exon V to generate Exon V–VIII–IX and V–VIII–VIIIh–IX transcripts [2]. It is worth noting that in publications before the year 2007, the present *BDNF* promoters I, II, IV, and VI were named Promoters I, II, III, and IV due to the order in which they were discovered [1]. In 2006, the distinct genomic structure of brain-derived neurotrophic factor (BDNF) was identified in humans, which included the discovery of a novel gene termed “*BDNFOS*” [2,3]. This gene is capable of producing alternatively spliced natural antisense transcripts (NATs). Several investigations have suggested that certain polymorphisms present in the *BDNFOS* gene are associated with specific disorders such as heroin addiction [4], lumbar disc herniation [5], and hypothalamus–pituitary axis regulation in major depression [6]. However, the molecular and physiological mechanisms underlying the functions of these NATs remain unclear.

### 1.2. The Expression of Bdnf Driven by Different Promoters

In general, these alternatively spliced transcripts, driven by distinct promoters, can produce only one protein, BDNF. However, the question remains as to why the expression of a single protein requires so many distinct promoters. Recently, a series of studies have indicated that *Bdnf* transcripts originating from different promoters can govern distinct cognitive and metabolic functions [7,8,9,10,11] (for a summary, see Table 1). The tissue-specific expression of *Bdnf* transcripts was examined comprehensively on the basis of quantitative or semiquantitative RT-PCR analyses [1,2,12]. In rodent brains, *Bdnf* transcripts driven by Promoters I, II, III, IV, VI, VIII, and IXA could be detected in nearly all brain regions, while Promoters V and VII appear to be applied specifically in the hippocampus and cortex [1]. In human brains, Promoters V and VII showed less obvious tissue-specificity compared with rodent brains [2]. Notably, the same brain region contains a vast range of neuronal types that are classified by morphological, anatomical, and physiological features.

To investigate the function of specific *Bdnf* promoters, our group (Dr. Bai Lu and colleagues) generated mouse lines, named *Bdnf*-e1^−/−^, e2^−/−^, e4^−/−^, and e6^−/−^ mice, in which *Bdnf* Promoters I, II, IV, and VI were selectively disrupted. Specifically, an enhanced GFP-STOP codon sequence was inserted downstream of a non-coding exon, resulting in the transcription of an exon-GFP-STOP codon-coding sequence RNA and the translation of enhanced GFP instead of *Bdnf* protein (Figure 2). These mouse lines can be used to investigate the function of *Bdnf*-specific promoters as well as to map the expression of neuron-specific *Bdnf* transcripts. So far, the neuronal expression of Exons I, II, and VI has been characterized [7,11,23]. *Bdnf*-e1 expression neurons (neurons expressing green fluorescence protein (GFP) driven by *Bdnf* Promoter I) were found in abundance in the cingulate cortex, lateral hypothalamus (LH), piriform cortex [13], basal lateral amygdala (BLA), medial amygdaloid nucleus (MePV), and some amygdala-adjacent cortex regions [7]. *Bdnf*-e6 neurons were mostly found in some hippocampal areas and multiple regions in the frontal cortex [11]. However, the functions of these groups of neurons expressing specific *Bdnf* transcripts remain largely elusive.

## 2. The Regulation of *Bdnf* Transcripts at the Molecular Levels

BDNF plays a crucial role in neuronal development and a wide range of neuronal functions, including neuronal survival, neurite outgrowth, and synaptic plasticity. The biochemical mechanisms underlying the downstream signaling of BDNF and its receptor TrkB have been extensively studied (see the reviews in [9,39,40,41,42,43]). More recently, emerging data on transcriptional regulation have improved our understanding of the complex functions of *Bdnf* in the neurons. Among the nine promoters of *Bdnf*, the regulatory mechanisms of Promoters I and IV have been the subject of extensive investigation at the molecular level over the past two decades [13,44,45,46,47,48,49,50].

### 2.1. Bdnf Promoter I

*Bdnf* Promoter I contains several enhancer or repressor elements, including a binding site for the nuclear factor kB (NF-kB) [51], the calcium response elements (CaREs) bound by CRE binding protein (CREB) [46], and neuron restrictive silencer elements (NRSE) bound by repressor element-1 silencing transcription factor (REST) [52,53,54] (Figure 3A). In primary neuronal cultures, the selective activation of 5-HT and norepinephrine receptors leads to an increase in *Bdnf* transcripts from Promoter I, mediated by cAMP, Ca^2+^, and CREB [48]. A recent study has shown a new enhancer for Promoters I, II, and III, located approximately 3 kb downstream of the Exon I transcription start site, which is probably regulated by CREB and AP-1 family proteins, as supported by a ChIP-seq analysis [49]. The transcription from *Bdnf* Promoter I can also be regulated by epigenetic factors. Hara et al. showed that potassium chloride (KCl) induced depolarization, leading to the acetylation of histones H3 and H4, mediated by Ca^2+^ signals, and resulting in a significant increase in the amount of Promoter I-driven *Bdnf* transcripts [54].

Tabuchi et al. [55] used transfection of the *Bdnf* Promoter I sequence linked to a reporter gene in primary neuronal cultures and found that the inducibility of Promoter I was confined to neuronal cultures, indicating that Promoter I may be neuron-specific. However, Promoter I-driven transcripts were also detected in the thymus [1], the function of which is still unknown. Further in vitro studies have shown that, unlike *Bdnf* Promoter VI, Promoters I, II, and IV exhibit activity dependence [46,48,54]. Interestingly, the expression of *Bdnf*-e1 transcripts can last for at least 6 h in the absence of depolarization [54]. Similar long-term expression of *Bdnf*-e1 transcripts was observed in the somatosensory cortex [56]. After unilateral whisker stimulation, *Bdnf*-e1 transcripts peaked at 8 h of stimulus and remained at a high level after 24 h [56]. In addition to its activity dependence, Promoter I also exhibits constitutional expression [7], but the molecular mechanisms underlying this mode of expression remain unknown.

The *Bdnf* gene produces alternative spliced transcripts that include a non-coding exon and a coding exon, Exon IX, which has an AUG translation start site located 21 base pairs after the splice site. It is noteworthy that most *Bdnf* transcripts initiate translation from this Exon IX start site. However, at the end of the non-coding Exon I, there is another in-frame AUG site that is used to synthesize pre-pro-*Bdnf* proteins that are eight amino-acid residues longer [57]. The function and physiological significance of this extended protein, which is generated by the *Bdnf*-e1 promoter, is not yet known (Figure 3B). 

As of now, the cellular functions of *Bdnf*-e1 transcripts are not well understood. However, a recent study conducted in mice indicated that the knockout of *Bdnf* transcripts from Promoter I in the piriform cortex resulted in a decrease in dendritic spines following the application of electroconvulsive therapy (ECT). This suggests a relationship between Promoter I and dendritic structural plasticity in the piriform neurons [13]. Further research is needed to elucidate the specific cellular functions of *Bdnf*-e1 transcripts in different types of neurons and under various physiological and pathological conditions.

### 2.2. Bdnf Promoter II

Similar to Promoter I, Promoter II is an activity-dependent promoter. Fewer molecular and cellular studies have been conducted on this promoter compared with those on Promoters I and IV [58,59]. Although the activity-responsive transcriptional factor and its binding site to Promoter II remain unknown, a unique feature of Promoter II is the presence of a neuron-restrictive silence element (NSE) [19,60], which can bind to the zinc finger protein neuron-restrictive silencer factor (NRSF) and be co-repressed by mSin3A and CoREST (for a review, see [50]) (Figure 3A). Interestingly, some findings have indicated that in injured neurons, the spinal cord injury and regeneration-related protein #69 (SCIRR69) can specifically bind to and activate *Bdnf* Promoter II, suggesting a potential role of this promoter as an endogenous target for neuronal repair or regeneration [19].

### 2.3. Bdnf Promoter IV

The *Bdnf* Promoter IV was initially classified as Promoter III until 2007, when several new *Bdnf* promoters were discovered, leading to a change in the nomenclature [1]. *Bdnf* Promoter IV contains multiple binding sites for transcriptional factors, including three calcium-responsive elements (CaRE) co-activated by the calcium-response factor (CaRF), USF1/USF2, and CREB [59,61,62,63]. Additionally, it has an NF-kB binding site [64] and an E-box region bound with the basic helix–loop–helix (BHLH) protein BHLHB2, a transcriptional repressive factor [65] (Figure 3A).

Epigenetic factors such as DNA methylation [63], histone methylation, and histone acetylation [31] can regulate the expression of Promoter IV in an activity-dependent manner. For instance, Chen et al. [63] found that methyl-CpG binding protein 2 (MeCP2) selectively binds to Promoter IV to repress the synthesis of *Bdnf*, while neuronal activity can trigger the phosphorylation and dissociation of MeCP2 from Promoter IV in the cortical neurons. To quantify the direct correlations between DNA methylation and distinct transcriptional factors, a newly developed method called AlphaScreen was preliminarily applied to analyze *Bdnf* Promoter IV and its CpG methylation [66]. This approach may reveal the actual contributions of epigenetic factors to the expression of Promoter IV in various neuronal functions and behaviors.

Similar to Promoter I, Promoter IV exhibits robust activity dependence in the neurons. Several studies have confirmed that depolarization via the application of KCl [61,62] or kainic acid [58,67] can trigger the expression of Promoter IV-driven transcripts in primary neuron cultures. 

During early postnatal development in the visual cortex of rats, in contrast to Promoters I and II, Promoters IV and VI were found to be expressed, as determined by an in situ hybridization assay [24]. Interestingly, Pattabiraman et al. [24] also demonstrated that *Bdnf*-e6 transcripts are present in both cell bodies and dendrites, while *Bdnf*-e4 transcripts are located exclusively in cell bodies. These findings suggested that Promoter VI-driven *Bdnf* may be involved in distal dendritic functions such as synaptic plasticity, while Promoter IV may be specific for maintaining cell survival and producing autocrine *Bdnf* to assist with the integration of cellular information in the neuronal soma.

Using *Bdnf*-e4^−/−^ mice, Sakata et al. [25] demonstrated that Promoter IV-driven *Bdnf* contributed to the development of PV interneurons, GABAergic transmission, and synaptic plasticity. Furthermore, *Bdnf*-e4^−/−^ mice exhibited reduced expression of specific serotonin receptors (1b, 2a, and 4b) and increased expression of the serotonin-synthesizing enzyme tryptophan hydroxylase (TPH) and the dopamine D4 receptor (DRD4) [68]. These results suggested that Promoter IV-driven *Bdnf* is involved in the regulation of monoamine systems. Additionally, *Bdnf*-e4^−/−^ mice displayed cholinergic dysfunctions in the frontal cortex and hippocampus compared with wild-type mice [69].

In addition to being expressed in the neurons, *Bdnf*-e4 transcripts have also been found in cultured rat cortical astrocytes following extracellular ATP stimulation [70]. Future in vivo studies of the expression of Promoter IV in astrocytes are necessary to confirm this phenomenon.

### 2.4. Promoter VI

Initially classified as Promoter IV, *Bdnf* Promoter VI underwent a change in nomenclature in 2007 when several new *Bdnf* promoters were discovered [1]. *Bdnf* Promoter VI contains multiple binding sites for transcriptional factors, including two GC boxes and a GA-rich sequence that can both be bound by specificity protein 1 (Sp1) [47]. It also contains a CCAAT box that is bound by CCAAT/enhancer-binding protein β (C/EBPβ) and regulated by Ca^2+^/calmodulin-dependent protein kinase II (CaMKII) [47] (Figure 3A).While corticosterone treatment can downregulate Promoter VI-driven transcripts in the hippocampus [71], NGF treatment can induce the expression of Promoter VI through the ERK1/2 pathway [72]. Interestingly, Promoter VI exhibited a fast transient expression similar to that of Promoter II, which was dependent on the acetylation of the histone H3K27 by CREB-p/CBP [73].

### 2.5. Summary

Among the four investigated *Bdnf* promoters, Promoters I, II, and IV exhibit activity-dependent characteristics. Promoters I and IV share similar regulatory elements, such as PasRE, NFkB-RE, and CRE, which can be activated by the transcriptional factors NPAS4, NFkB, and CREB, respectively (Figure 3A). In addition to these elements, Promoter IV also has specific regulatory elements, including CaRE1 and BHLHB2-RE, which contribute to a different distribution from Promoter I-driven transcripts. It seems that compared with Promoter I, Promoter IV is more sensitive to a rise in the intracellular Ca^2+^ concentration, and could be activated by an influx of Ca^2+^ via not only the L-type Ca^2+^ channel but also NMDA receptor. Promoter I and II are located in the same cluster and are quite close (558 bp) in the genomic structure of *Bdnf*, so Promoter II may also use the regulatory elements upstream of Promoter I. However, it also exhibits a specific regulatory element downstream of its own transcription start site (TSS), the NRSE. Interestingly, Exon II contains two more alternative splice sites, leading to three different transcripts from Promoter II (IIa, IIb, and IIc). However, the specific functions of Transcripts IIa, IIb, and IIc are still unknown. Distinct from Promoters I, II, and IV, Promoter VI has different regulatory elements, such as GC box and CCAAT sequences, which are not responsive to activity.

Although the cellular functions of Promoter IV-driven BDNF have been discovered to contribute to the development of PV interneurons, GABAergic transmission, and synaptic plasticity [25], the functions of the other transcripts at the cellular level are still elusive. mRNA localization analysis may help us to understand the underlying mechanisms. Interestingly, while BDNF is well known for being expressed by cortical and hippocampal glutamatergic neurons, a series of studies have shown that other cells, such as microglia [74,75,76,77,78] and GABAergic neurons [79], could also express BDNF. Microglia-released BDNF was found to contribute to neuropathic pain [75,76], learning-dependent synapse formation [77], and activation of the microglia themselves [78]. To date, the promoters driving the expression of BDNF in the microglia are still unknown. 

Recent research has also demonstrated that various metal ions, such as copper, zinc, and lithium, play a significant role in modulating the expression and functions of BDNF [80]. Zinc and copper have been observed to induce conformational changes through direct interactions with the neurontrophins, thereby influencing the recognition processes of their receptors [81,82,83,84,85,86]. Additionally, copper [87,88] and lithium [89,90] have been shown to act as activators for the expression of BDNF. However, the precise mechanisms by which metal ions regulate the expression and release of BDNF are not yet fully understood. Further investigations are required to determine whether specific BDNF promoters can be induced by different metal ions. Such research could provide valuable insights into the roles of metal ions in the regulation of BDNF.

## 3. The Function of *Bdnf* Transcripts in Psychiatric Disorders and Cognitive Functions

*Bdnf* is known to play a critical role in a wide range of cognitive functions and has been implicated in various psychiatric and neurological disorders, such as depression, anxiety, schizophrenia, and neurodegenerative diseases. Over the past decade, numerous reviews have provided comprehensive summary and discussion of this topic [8,91,92,93]. In this section, we review and discuss the findings on specific *Bdnf* transcripts/promoters in different psychiatric disorders and cognitive functions. 

### 3.1. Depression and Bdnf Promoter IV

Over the past three decades, extensive research has explored the relationship between *Bdnf* and mood disorders, particularly depression and anxiety (for a comprehensive review, see [8]). Despite this extensive body of research, several key questions remain unanswered. One critical question is whether dysfunctional *Bdnf* genes contribute to the development of depression, or whether depression itself causes a reduction in the expression of *Bdnf*, or if both factors play a role. In addition, there is a need to investigate the potential of the *Bdnf*–TrkB pathway as a target for treating major depression, and to examine the role of *Bdnf* during antidepressant treatment. Exploring the effects of *Bdnf* promoters on these questions may offer a new perspective for future research.

A human study has indicated that the *BDNF* Promoter IV may play a role in the development of depression [26]. To be specific, a *BDNF* gene polymorphism known as the C allele at rs12273363 has been found to be associated with mood disorders in humans, as compared with the more common T allele [26]. Notably, Hing et al. discovered that this polymorphic site is located within an evolutionarily conserved region (ECR) called EC5.2, which is a potent and specific repressor of Promoter IV-mediated activity-dependent *Bdnf* transcription [27]. These findings suggest a potential link between the transcription of Promoter IV and depression in humans.

Social defeat is a widely used behavioral paradigm to induce depressive behaviors in mice. Tsankova et al. [31] found that after social defeat, the *Bdnf* transcripts driven by Promoters IV and VI were significantly reduced, while the transcripts driven by Promoters I and II remained unaffected. This reduction could be rescued by chronic treatment with imipramine, a tricyclic antidepressant [31]. However, whether *Bdnf* alone, without the contribution of traditional antidepressants, can rescue depression in socially defeated mice is an unexplored question. To investigate whether dysfunction in the *Bdnf* Promoter IV gives rise to depression, Sakata et al. [28] conducted depression-like behavioral tests on *Bdnf*-e4^−/−^ mice. The study found increased immobility time in the tail suspension test (TST), less preference for sucrose, and elevated escape latency in the learned helplessness test (LHT) [28]. In another study, antidepressants were found to rescue the depressive phenotype in *Bdnf*-e4-/- mice, but with no increase in hippocampal *Bdnf* levels [29]. While this was interpreted as though antidepressants could function independently of *Bdnf*, one cannot rule out the possibility that the antidepressants work through other *Bdnf* promoters or in brain regions other than the hippocampus.

Interestingly, providing an enriched environment to *Bdnf*-e4^−/−^ mice at an early age induced the secretion of *Bdnf* in the hippocampus and rescued depressive-like behaviors [30]. Thus, other *Bdnf* promoters have been implicated in depressive disorders. Further work studying the relationship between early brain development and *Bdnf* promoters may reveal the underlying mechanisms.

The complex relationship between antidepressants and the expression of *Bdnf* may not be attributed solely to Promoter IV but also to other *Bdnf* promoters. Some findings have suggested that different antidepressants can activate distinct *Bdnf* promoters. Specifically, Tsankova et al. demonstrated that imipramine activated Promoters IV and VI, but not I and II [31], whereas Dias et al. [20] observed that chronic administration of tranylcypromine enhanced the *Bdnf*-e2 transcripts, while desipramine treatment increased *Bdnf*-e4 transcripts specifically. However, fluoxetine treatment did not significantly affect all *Bdnf* transcripts [20]. Moreover, Russo-Neustadt et al. showed that tranylcypromine treatment combined with voluntary physical activity can induce *Bdnf*-e1, but not the production of e2 in the rat hippocampus [14]. Together, the influence of antidepressants on the expression of *Bdnf* is diverse and complex.

In a human study, Tadić et al. [32] provided preliminary evidence that antidepressants may be ineffective for people exhibiting hypomethylation in Promoter IV. Similarly, another study found that hypomethylation of the CpG-87 site located in the Promoter IV region was associated with reduced antidepressant efficacy in patients with major depressive disorder (MDD) [33]. These findings suggested that antidepressants may compete with the Promoter IV-driven expression of *BDNF* in the treatment of MDD. A recent cell culture study reported that fluoxetine and venlafaxine could increase the methylation of *BDNF* Promoter IV, and the methylation of position CpG-39 could significantly elevate the activity of Promoter IV [34]. Interestingly, a recent human study reported that tobacco use is strongly correlated with the methylation of *BDNF* Exon IV in depressed individuals [36]. Further studies are needed to elucidate the complex interplay between antidepressants and the expression of *Bdnf*, both dependent and independent of Promoter IV.

### 3.2. Schizophrenia and Bdnf Promoter VI

BDNF protein is highly expressed in the brain regions implicated in schizophrenia, such as the prefrontal cortex, parietal cortex, and hippocampus. Several studies have investigated the relationship between *BDNF* and schizophrenia, and have identified a well-known *BDNF* polymorphism, rs6265 (C→T, Val/Met), as a potential biomarker for the risk of developing schizophrenia (see [92,94] for a review). At the transcriptional level, multiple studies have shown that schizophrenia is associated with specific *BDNF* promoters [11,21,95].

In a human study of cohorts of schizophrenia patients in the United States and Australia, Wong et al. [21] found that the transcripts of both *BDNF*-e2 and -e6 were significantly reduced in the parietal cortex, *BDNF*-e2 transcripts were reduced in the dorsal lateral prefrontal cortex (DLPFC), and *BDNF*-e6 transcripts were decreased in the hippocampus of schizophrenia patients compared with healthy controls. In another study, Chen et al. [11] used *Bdnf*-e6^−/−^ mice to investigate the relationship between *Bdnf* Promoter VI and schizophrenia-like phenotypes. They found that a deficiency in the Promoter VI-driven expression of *Bdnf* combined with adversity early in life gave rise to schizophrenia-like behaviors, such as deficits in social interactions, spatial memory, and sensorimotor gating, as measured by the pre-pulse inhibition test (PPI) [11], indicating a direct relationship between *Bdnf* Promoter VI and the risk of developing schizophrenia-like behavior, at least in rodents. 

### 3.3. Anxiety

Chen et al. [96] identified a significant role for BDNF in anxiety-related behaviors by demonstrating that the *BDNF* single nucleotide polymorphism rs6265 (C→T, Val/Met) was involved in anxiety-related behaviors. *Bdnf* ^Met/Met^ or *Bdnf*
^+/–^ mice exhibited anxiety-like behaviors that could not be rescued by the antidepressant fluoxetine. Several subsequent studies have further explored the relationship between BDNF and anxiety, including the reviews by [8,97,98].

Sakata et al. reported that *Bdnf*-e4^−/−^ mice showed reduced activity in the open field test (OFT), reduced food intake in the novelty-suppressed feeding test (NSFT), and defective response inhibition in the passive avoidance test (PAT) [28], which could not be rescued by phenelzine [29], a monoamine oxidase inhibitor. However, *Bdnf*-e4^−/−^ mice did not exhibit anxiety-like behavior in the light and dark box test (LDB) and elevated plus maze test (EPM) [28], two reliably and frequently used tests for anxiolytics [99]. These findings suggested that the disruption specifically of Promoter IV may not lead to anxiety-like behavior, unlike the case of *Bdnf*^+/–^ mice. It is possible that defects in other promoters may contribute to anxiety in specific situations. Further research is needed to clarify the mechanisms underlying the relationship between BDNF and anxiety and to identify the specific promoters involved. 

### 3.4. Cocaine Addiction and Bdnf Promoters I and IV

BDNF plays a critical role in cocaine addiction and withdrawal, which are governed by the reward system in the brain (see [3,100,101,102] for reviews). This system comprises several limbic sites, including the ventral tegmental area (VTA), nucleus accumbens (NAc), and medial prefrontal cortex (mPFC). The expression of BDNF was elevated in the mPFC following both a single dose of cocaine [103] and repeated cocaine exposure [104], while overexpression in the NAc and VTA was only observed following repeated exposure [104]. Moreover, BDNF induced by cocaine withdrawal facilitated LTP in the mPFC by suppressing the inhibition of GABA [105]. These findings suggested that BDNF may play multiple roles in cocaine addiction, including drug responses, the formation of addiction, withdrawal effects, and reward seeking. It is plausible that specific promoters may contribute to the distinct functions of BDNF in cocaine addiction.

A few studies have identified the specific functions of *Bdnf* promoters in addiction. Schmidt et al. [15] found that rats undergoing one week of cocaine withdrawal showed a significant increase in *Bdnf* Promoter I-driven transcripts in the VTA, but not in transcripts driven by Promoters II, IV, VI, or IX. This elevation in *Bdnf*-e1 transcripts was associated with the specific binding of CREB to Exon I and the acetylation of Histone 3 in Exon I [15]. Additionally, spontaneous morphine withdrawal, as well as cocaine withdrawal, led to increased *Bdnf*-e1 transcripts in the frontal cortex and midbrain [16]. Interestingly, unlike a single dose of cocaine [104], acute morphine intoxication did not influence all *Bdnf* transcripts in multiple brain regions [16], suggesting differences in the responses and downstream mechanisms of *Bdnf* after the application of different addictive drugs.

As *Bdnf* Promoter IV is susceptible to epigenetic modulation by addictive drugs, it is believed to be another promoter involved in the addiction and reward circuitry [106]. Notably, Tian et al. [17] observed that *Bdnf*-e4 transcripts were upregulated exclusively in the nucleus accumbens (NAc) of mice conditioned to cocaine, with no such increase observed in mice treated with cocaine but not conditioned to it. Interestingly, *Bdnf*-e1 transcripts were found to respond to both cocaine treatment and conditioning [17], suggesting that while Promoter IV may be involved in cocaine-related reward-seeking behavior, Promoter I responds to the direct pharmacological effects of cocaine.

Furthermore, nicotine is another addictive chemical that impacts the health of millions of smokers. Two recent human studies conducted in Germany Germany [35] and Brazil [36] reported dysregulated methylation patterns in *BDNF* Exon IV among smokers. Future studies investigating the specific mechanisms through which nicotine shapes the methylation patterns in Exon IV could help shed light on the mechanism of nicotine addiction.

### 3.5. Aggression

Hemizygous deletions or mutations of the *BDNF* gene in humans have been associated with various psychiatric phenotypes, including anxiety, mood disorders, and aggressive behaviors (see [107] for a review). In addition, heterozygous *Bdnf*^+/-^ mice exhibited enhanced intermale aggressiveness [108]. The study by Maynard et al. has investigated the contribution of specific *Bdnf* promoters to aggressive behavior [10]. They found that male *Bdnf*-e1^−/−^ and e2^−/−^ mice displayed increased aggressive behavior compared with their wild-type littermates [10]. Interestingly, a recent study found higher expression levels of Exons I, II, III, V, VII, and VIII in the cortex of aggressive rats compared with tame ones [109], indicating that multiple *Bdnf* promoters may contribute to aggressive behaviors.

### 3.6. Rett Syndrome and Promoter IV

Rett syndrome (RTT) is a neurodevelopmental disorder caused by loss-of-function mutations in the gene encoding methyl-CpG-binding protein 2 (MeCP2) [110], which is a transcriptional regulatory protein. It affects approximately 1 in 10,000 live female births worldwide and is characterized by normal early postnatal development with neurological symptoms that appear around 6–18 months of age (see [111,112,113] for a review). The dysregulation of BDNF has been observed in RTT patients and mouse models of the disease, and has gained attention in the search for potential therapies for RTT and related syndromes [110,111,112,113,114,115,116,117]. The discovery of dysregulated expression of BDNF in RTT and mouse models of the disease has spurred progress in developing BDNF-targeted therapies for the treatment of RTT (see [112] for a review). For example, the application of the BDNF-inducing drug fingolimod, which is an agonist of the sphingosine-1-phosphate (S1P) receptor, may prevent the abnormalities in the neuronal structure that are typical of RTT [117]. 

Several studies have revealed a relationship between MeCP2 and *BDNF* Promoter IV [63,118,119]. Specifically, previous studies have demonstrated the binding of the MeCP2 protein to the *BDNF* Promoter IV by using chromatin immunoprecipitation [63,118]. Notably, strong chemical-induced depolarization of neurons can reduce the binding of the MeCP2 protein, accompanied by a shift in the phosphorylation of MeCP2 [63]. In addition, Zocchi and Sassone-Corsi et al. found that mice lacking functional Sirtuin 1 (SIRT1), which is a nicotinamide-adenine dinucleotide-dependent histone deacetylase, exhibited increased MeCP2 binding to the *BDNF* Exon IV promoter and decreased levels of BDNF mRNA and protein [119]. The relevance of these findings to human patients remains to be tested clinically. 

### 3.7. Some Other Cognitive Functions Related to Specific Bdnf Promoters

Previous studies have shown that specific *Bdnf* promoters are involved in various brain functions, such as circadian rhythm, sleep, and memory formation. For instance, Berchtold et al. [22] found that the levels of transcripts driven by Promoter II and IV fluctuated during a diurnal cycle. To investigate whether the changes in the expression of Promoter II or IV contributed to circadian rhythms or are merely a response to light–dark stimuli, future studies may need to examine the circadian rhythm in *Bdnf*-e1*^−/−^* and e2^−/−^ mice. Additionally, BDNF plays a critical role in sleep regulation and is influenced by sleep deprivation [120]. Hill et al. observed that *Bdnf*-e4^−/−^ mice exhibited a decrease in slow-wave activity during non-rapid eye movement (NREM) sleep [37], indicating a dysfunction in sleep homeostasis. Furthermore, BDNF is involved in long-term memory and fear-conditioned memory (see [98,121,122,123] for reviews). To be specific, *Bdnf* Promoter IV appears to control executive functions related to learning and memory. For example, Sakata et al. found that *Bdnf*-e4^−/−^ mice exhibited impaired spatial memory reversal and contextual memory extinction, which are two executive functions, but retained their spatial memory and contextual learning capacity [38].

## 4. The Function of *Bdnf* Transcripts in Different Aspects of Metabolism

In mammals, the regulation of energy homeostasis is governed by the balance between energy intake (food consumption) and energy expenditure, which includes basal metabolism, physical activity, adaptive thermogenesis, and food digestion [124]. Several studies have highlighted the role of BDNF-TrkB signaling in the regulation of body weight and metabolism in humans [125,126,127]. For instance, research has shown that *Bdnf* haploinsufficiency in children with Wilms’ tumor, aniridia, genitourinary anomalies, and mental retardation (WAGR) syndrome is associated with severe early-onset obesity [125].

BDNF is primarily expressed in the central nervous system, and its regulation of peripheral metabolic function is thought to occur via neuronal projections from the brain. Numerous studies have shown that BDNF-expressing neurons, predominantly located in the nuclei of the hypothalamus and medulla, play a crucial role in the maintenance of the energy balance [128]. One particularly noteworthy study found that distinct subpopulations of BDNF-expressing neurons in the paraventricular hypothalamus were involved in the regulation of various aspects of energy homeostasis, including food intake and thermogenesis [129]. These findings highlighted the multifaceted functions of BDNF in the regulation of the energy balance.

It is plausible that different BDNF transcripts may contribute to different aspects of the energy balance. Our group has developed *Bdnf*-e1^−/−^, e2^−/−^, e4^−/−^, and e6^−/−^ mice, in which *Bdnf* Promoters I, II, IV, and VI were selectively mutated [10,25,38]. In a study characterizing the metabolic functions of these mice [18], *Bdnf*-e1^−/−^, e2^−/−^ mice were overweight, while *Bdnf*-e4^−/−^, e6^−/−^ mice maintained a normal weight, consistent with the findings from another study [7]. Interestingly, McAllan et al. observed a hyperphagia phenotype in *Bdnf*-e1^−/−^ mice housed in a cage with a transparent divider to avoid aggressive attacks [18]. In contrast, You et al. [7] and Chu et al. [23] found no hyperphagia in group-housed *Bdnf*-e1^−/−^ mice under normal conditions. It is possible that partial social isolation in the context of *Bdnf*-e1 knockout, but not *Bdnf*-e1 knockout alone, may lead to hyperphagia.

Moreover, a deficit in thermogenesis was observed in *Bdnf*-e1^−/−^ mice, and the *Bdnf*-e1-expressing neurons in the lateral hypothalamus (LH) were found to specifically regulate thermogenesis [7]. Additionally, Chu et al. reported that *Bdnf*-e2^−/−^ mice exhibited significantly increased food intake, indicating that *Bdnf* Promoter II, rather than Promoter I, may be involved in the regulation of hyperphagia. They also identified a group of *Bdnf*-e2 neurons in the ventral medial hypothalamus (VMH) that are crucial for promoting satiety. Based on the studies by You et al. [7] and Chu et al. [23], it can be hypothesized that under non-stress conditions, BDNF driven by Promoters I and II may contribute to distinct aspects of the energy balance through distinct groups of neurons in the hypothalamus (Figure 4). These findings provide preliminary evidence for the relationship between *Bdnf* promoters and different aspects of the energy balance, and further investigations focusing on the functions of other promoters in the energy balance would be valuable.

## 5. Perspectives for Future Research on *Bdnf* Promoters

In this article, we have provided a comprehensive overview of the molecular regulations and structural features of multiple *Bdnf* promoters, as well as a summary of the current knowledge on the cellular and physiological functions of different *Bdnf* transcripts derived from these promoters. 

Despite the significant progress made in this field, several important questions remain to be addressed. First, the mechanism underlying the selective expression of different *Bdnf* promoters in specific brain regions or neurons is not yet fully understood. Although some studies have identified that specific brain regions [1,2,12] or neurons [7,11,13] can express distinct *Bdnf* promoters, further investigation is required to elucidate the precise mechanisms responsible for the choice of promoter. Second, the spatial distribution of *Bdnf* transcripts and their corresponding functions at the cellular level is another important question that needs to be addressed. Although single-cell transcriptomic analyses and emerging bioinformatic tools have provided a more comprehensive map of the distribution of *Bdnf* transcripts, the underlying regulatory networks are still difficult to interpret without the aid of mathematical modeling. Third, understanding the downstream effects of the expression of specific *Bdnf* promoters in the neurons and how these promoters contribute to distinct neuronal circuits is also an area of active research. The development of newly engineered genetic lines with specific *Bdnf* promoter-driven Cre proteins, in conjunction with optogenetic tools and tracing viruses, is expected to provide new insights into the neuronal circuits downstream of the expression of specific *Bdnf* promoters. Finally, the temporal regulation of the expression of BDNF and the potential role of different promoters in brain development is another area of research that requires further investigation. The development of BDNF sensors coupled with genetically designed *Bdnf* promoter-driven tools may enable the visualization of the expression and release of BDNF controlled by different promoters during brain development.

In summary, the study of *Bdnf* non-coding promoters has the potential to reveal new insights into the complexity of BDNF’s function and provide novel therapeutic targets for the treatment of neurological disorders. Further investigations are required to fully understand the mechanisms underlying the choice of promoter, the spatial distribution, the downstream effects, and the temporal regulation of the expression of BDNF.

## Figures and Tables

**Figure 1 biomolecules-13-00655-f001:**
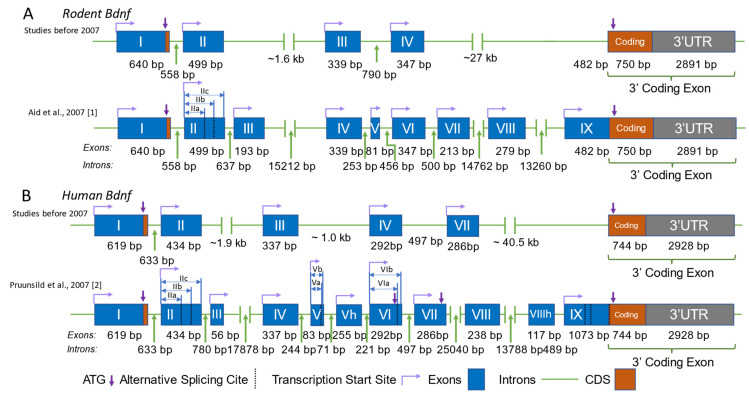
Genomic structure of BDNF in humans and rodents. Comparison of the genomic structure of the *Bdnf* gene in rodents (**A**) and humans (**B**), as proposed by studies before and after 2007. The structures presented are according to [1,2] The blue boxes represent exons, the green lines represent introns, the purple arrows indicate the translation start site (ATG), the dashed black lines indicate alternative splice sites, and the bent lines with arrows represent the transcription start site. The lengths of the exons and introns are labeled accordingly. Identical human exons and the rodent exons homologous to the human exons are named with the same number in all panels.

**Figure 2 biomolecules-13-00655-f002:**
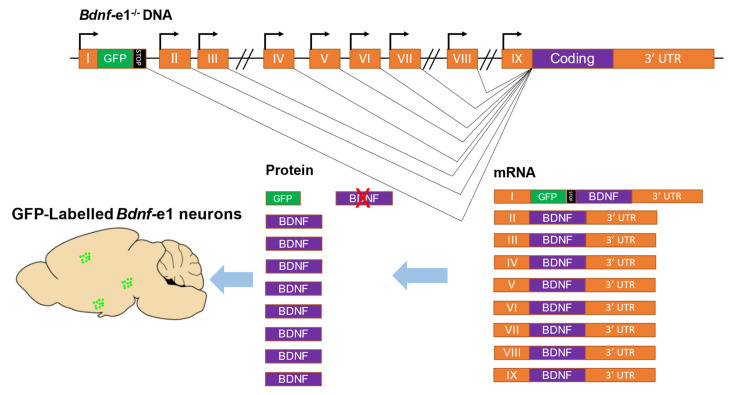
Strategy for generating mice with disruptions of specific *Bdnf* transcripts. As an example, the disruption of transcripts driven by Promoter I in mice is shown. The rodent *Bdnf* gene contains nine promoters that drive nine exons (I to IX). Exons can be transcribed from their respective promoters and alternatively spliced to the coding region to translate the same brain-derived neurotrophic factor (BDNF) protein. To generate *Bdnf*-e1^−/−^ mutant mice, a GFP-STOP cassette (the GFP sequence followed by multiple stop codons) was inserted after Exon I. In these mice, transcripts driven by Promoter I express GFP, which can be used as a marker for labeling *Bdnf*-e1 neurons, but not BDNF protein due to the presence of stop codons. The orange boxes represent exons, the black lines represent introns, the green and black boxes represent the GFP-STOP cassette, and the bent lines with arrows represent the transcription start site, and the green dots represent *Bdnf*-e1 expressing neurons.

**Figure 3 biomolecules-13-00655-f003:**
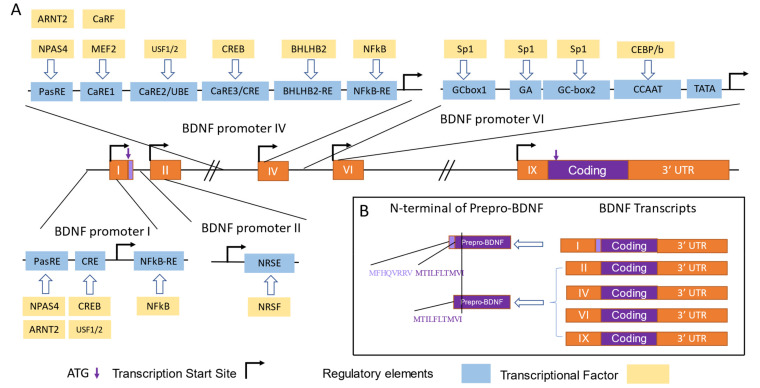
Translational and transcriptional control of BDNF by Promoters I, II, IV, and VI. (**A**) Transcriptional factors (TFs) and regulatory elements involved in the regulation of activity-dependent transcription from BDNF Promoters I, II, IV, and VI are depicted according to [2,47,50]. The orange boxes represent exons, the black lines represent introns, the bent lines with arrows represent the transcription start site, the purple arrows indicate the translation start site (ATG), the blue boxes represent regulatory elements, the yellow boxes represent TFs, the purple boxes represent Bdnf coding sequences or pre-pro-BDNF, and the light purple boxes represent the extended N-terminus of pre-pro-BDNF translated by Promoter I-driven transcripts. The abbreviations of the TFs and the regulatory elements are as follows. For Promoter I, neuronal PAS domain protein 4 (NPAS4) and aryl hydrocarbon receptor nuclear translocator 2 (ARNT2) bind with the basic helix–loop–helix PAS transcription factor response element (PasRE), nuclear factor kappa B (NFκB) binds with NFκB-RE, and cAMP/Ca^2+^-response element binding protein (CREB) and upstream stimulatory Factors 1 and 2 (USF1/2) bind with CRE-like element (CRE). For Promoter II, neuron-restrictive silencing factor (NRSF) binds with neuron-restrictive silencing element (NRSE). For Promoter IV, NPAS4 and ARNT2 bind with PasRE; Ca^2+^-response factor (CaRF) and myocyte enhancer factor-2(MEF2 bind with Ca^2+^-response element 1 (CaRE1); CREB binds with CRE; helix–loop–helix domain-containing, Class B2 (BHLHB2) binds with BHLHB2-RE; and NFκB binds with NFκB-RE. For Promoter VI, Sp1 binds with GCbox 1, GCbox2, and the GA sequence, and CCAAT enhancer binding protein b (CEBP/b) binds with the CCAAT sequence. (**B**) Amino acid sequences of different pre-pro-BDNF N-termini are shown, with the amino acids encoded by transcripts driven by Promoters I, II, IV, and VI listed adjacent to the N-terminal sequences.

**Figure 4 biomolecules-13-00655-f004:**
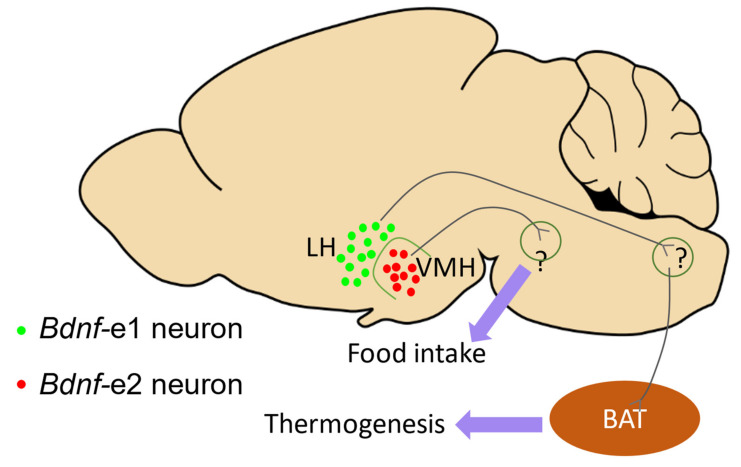
The function of BDNF driven by Promoters I and II in the energy balance. This diagram illustrates the hypothesis that BDNF driven by Promoters I and II may contribute to distinct aspects of the energy balance, according to studies by [7,23]. The distribution of *Bdnf*-e1 (green dots) and *Bdnf*-e2 (red dots) neurons in the lateral hypothalamus (LH) and ventral medial hypothalamus (VMH), and their potential roles in energy balance are shown. *Bdnf*-e1 neurons in LH are suggested to indirectly regulate brown adipose tissue (BAT, brown ellipse) and contribute to thermogenesis, while *Bdnf*-e2 neurons in VMH are postulated to control satiety.

**Table 1 biomolecules-13-00655-t001:** Summary of the functions of *Bdnf* Promoters I, II, IV, and VI, and references.

Promoter	Cellular, Cognitive, and Other functions	References
**I**	Dendritic spines formation	[13]
	Responses to antidepressants (tranylcypromine)	[14]
	Relations with cocaine withdraw	[15]
	Relations with spontaneous morphine withdraw	[16]
	Responses to cocaine	[17]
	Aggressive behavior	[10]
	Obesity	[7,18]
	Energy balance-thermogenesis	[7]
**II**	Spinal cord injury repairment	[19]
	Responses to antidepressants (desipramine)	[20]
	Relations with schizophrenia patients	[21]
	Aggressive behavior	[10]
	Circadian rhythm	[22]
	Obesity	[18,23]
	Energy balance-hyperphagia	[23]
**IV**	Cell body localization; Cell survival; Postsynaptic functions	[24]
	GABAergic transmission; synaptic plasticity	[25]
	The development of depression in human beings	[26,27]
	The function underlying depressive function in mouse model	[28,29,30]
	Response to antidepressants	[20,31,32,33,34]
	Anxiety-like behaviors	[28,29]
	Cocaine reward-seeking behavior	[17]
	Relations with nicotine addiction in human beings	[35,36]
	Circadian rhythm	[22]
	Sleep regulation and NREM sleep	[37]
**VI**	Spatial memory reversal and Contextual memory extinction	[38]
	Dendrites localization and Distal dendritic development	[24]
	Relations with schizophrenia patients	[21]
	Schizophrenia-like phenotypes in mice	[11]

## Data Availability

No data were created in this review paper.

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
