# Peer review of "Diverse Functions of Multiple Bdnf Transcripts Driven by Distinct Bdnf Promoters"

_biomolecules, 2023, doi:10.3390/biom13040655_

Round 1

Reviewer 1 Report

In their review “Diverse functions of multiple Bdnf transcripts driven by distinct Bdnf promoters” authors provide comprehenisve description of the structure and transcriptional regulation of the gene BDNF encoding brain-derived neurotrophic factor – secreted protein with multiple functions in neural tissues and beyond. In the current review they focus on distinct functions of transcripts driven by different promoters of human BDNF. They also provide brief overview of BDNF structure in other organisms (mice).

While there are several reviews have been published about role of the transcripts from different promoters of BDNF in particular physiological or pathological conditions, the current work is an attempt to summarise them all. Additionally, authors present and discuss their own recent data on this subject.

Thus, this review is of definite interest.

Please find below my comments and suggestions -

Should the name of the human BDNF gene be written using capital letters and italicized?

The authors may want to mention another recent article about BDNF exon IV and depression - Quaioto BR, Borçoi AR, Mendes SO, Doblas PC, dos Santos Vieira T, Moreno IA, Dos Santos JG, Hollais AW, Olinda AS, de Souza ML, Freitas FV. Tobacco use modify exon IV BDNF gene methylation levels in depression. Journal of Psychiatric Research. 2023 Mar 1;159:240-8.

Line 71. I suggest that instead of “To investigate the function of specific Bdnf promoters, Dr. Bai Lu and colleagues” the authors may say “To investigate the function of specific Bdnf promoters, our group (Dr. Bai Lu and colleagues)”

The Table 1 while being very informative is not easy to read, because of the small and rather fuzzy font of the text. I recommend authors to fix it, and also to replace red text in the heading with black or blue.

I also suggest to tone down the title/heading of the Table 1, as it’s not a summary of all BDNF specific promoter related functions and references. Also, the name of the gene (BDNF) is imssing in the heading of the Table 1.

Line 79. “Chu et al” - year is missing.

Line 185. The authors state “Interestingly, recent findings indicate….”, however publication from 2013 is not a recent finding. There are no recent articles on the role of CIRR69 in SCI. Are there other lines of evidence supporting specific role of promoter II-derived transcripts in neuronal repair/regeneration? Is it possible to speculate that impact of the promoter II-derived transcripts on energy balance can be somehow linked to this specific role?

Should the authors add into the article discussion about BDNFOS (so called anti-BDNF)-derived transcripts?

The idea about drug development targeting specific BDNF promoters is very appealing; can the authors suggest some methodological approaches to achieve such goal?

Do the authors have any data about computationally predicted regulators of specific promoter-driven BDNF transcription (in silico data, yet to be tested in vivo)?

Author Response

[Please also see the attachment.]

Reviewer #1

In their review “Diverse functions of multiple Bdnf transcripts driven by distinct Bdnf promoters” authors provide comprehensive description of the structure and transcriptional regulation of the gene BDNF encoding brain-derived neurotrophic factor – secreted protein with multiple functions in neural tissues and beyond. In the current review they focus on distinct functions of transcripts driven by different promoters of human BDNF. They also provide brief overview of BDNF structure in other organisms (mice). While there are several reviews have been published about role of the transcripts from different promoters of BDNF in particular physiological or pathological conditions, the current work is an attempt to summaries them all. Additionally, authors present and discuss their own recent data on this subject. Thus, this review is of definite interest.

Thank you for your review of our manuscript. We are glad that you found our review to be comprehensive and of definite interest and appreciate your recognition of our efforts to summarize the different functions of transcripts driven by distinct Bdnf promoters. We have made the necessary revisions to address your suggestions, and we hope that our updated manuscript meets your expectations. Thank you again for your positive feedback, and we hope that our manuscript contributes to the field of BDNF research.

Please find below my comments and suggestions -
1. Should the name of the human BDNF gene be written using capital letters and italicized?

The human BDNF gene should be written using capital letters and italicized as 'BDNF.' We have followed the guidelines for rodent [1,2] and human gene nomenclature [3], which suggest using the first capital letter with following lower-case letters for rodent genes (for example, 'Bdnf') and all capital letters for human genes (for example, 'BDNF').

2. The authors may want to mention another recent article about BDNF exon IV and depression - Quaioto BR, Borçoi AR, Mendes SO, Doblas PC, dos Santos Vieira T, Moreno IA, Dos Santos JG, Hollais AW, Olinda AS, de Souza ML, Freitas FV. Tobacco use modify exon IV BDNF gene methylation levels in depression. Journal of Psychiatric Research. 2023 Mar 1;159:240-8.

We have cited the recent article about BDNF exon IV and depression by Quaioto et al. 2023 in section 3.1 (lines 343-345).

3. Line 71. I suggest that instead of “To investigate the function of specific Bdnf promoters, Dr. Bai Lu and colleagues” the authors may say “To investigate the function of specific Bdnf promoters, our group (Dr. Bai Lu and colleagues)”

We have revised the sentence in line 71 according to the reviewer's suggestion by replacing "Dr. Bai Lu and colleagues" with "our group (Dr. Bai Lu and colleagues)".

4. The Table 1 while being very informative is not easy to read, because of the small and rather fuzzy font of the text. I recommend authors to fix it, and also to replace red text in the heading with black or blue. I also suggest to tone down the title/heading of the Table 1, as it’s not a summary of all BDNF specific promoter related functions and references. Also, the name of the gene (BDNF) is missing in the heading of the Table 1.

We have made several changes to Table 1 based on the reviewer's feedback. We have increased the font size, changed the font type to Times New Roman, and replaced the red text in the heading with black. Moreover, we have toned down the title/heading of the table and included the gene name 'BDNF' in the heading. The new heading is "Summary of Bdnf promoters I, II, IV and VI related functions and references."

5. Line 79. “Chu et al” - year is missing.

We have added the year 2023 to the citation of Chu et al.

6. Line 185. The authors state “Interestingly, recent findings indicate….”, however publication from 2013 is not a recent finding. There are no recent articles on the role of CIRR69 in SCI.

We have modified the sentence by replacing "recent findings indicate" with "some findings indicate" as the publication from 2013 is not a recent finding.

7. Are there other lines of evidence supporting specific role of promoter II-derived transcripts in neuronal repair/regeneration? Is it possible to speculate that impact of the promoter II-derived transcripts on energy balance can be somehow linked to this specific role?

There is currently no other evidence supporting the specific role of promoter II-derived transcripts in neuronal repair/regeneration in published papers. However, it is possible to speculate that the impact of promoter II-derived transcripts on energy balance can be linked to this specific role. Future studies may focus more on the relationship between brain injury or neurodegenerative diseases and promoter II-expressing neurons to test this speculation.

8. Should the authors add into the article discussion about BDNFOS (so called anti-BDNF)-derived transcripts?

Thanks for the suggestion. We have added a paragraph discussing BDNFOS in section 1.1.

9. The idea about drug development targeting specific BDNF promoters is very appealing; can the authors suggest some methodological approaches to achieve such goal?

Although targeting specific BDNF promoters directly may not be a potential method for drug development, specific BDNF promoter-expressing neurons could be a potential target for certain diseases. For instance, with future comprehensive single-cell analysis of promoter I-expressing neurons, their specific receptors could be discovered, and corresponding agonists could be selected. We have removed the sentence "Our analysis highlights the complexity of BDNF functions and emphasizes the potential for drug development targeting specific Bdnf promoters." in section 5, as the methodological approach has not been proven to work or not.

10. Do the authors have any data about computationally predicted regulators of specific promoter-driven BDNF transcription (in silico data, yet to be tested in vivo)?

Thanks for your wonderful idea. We have no data about computationally predicted regulators of specific promoter-driven BDNF transcription yet. The prediction of the binding of promoter DNA and transcriptional factors is usually inaccurate and not reliable. However, with emerging tools in machine learning and deep learning algorithms, we believe that in the near future, a vast number of specific regulators would be found in silico.

References

  1. Maltais, L.J.; Blake, J.A.; Eppig, J.T.; Davisson, M.T. Rules and guidelines for mouse gene nomenclature: a condensed version. International Committee on Standardized Genetic Nomenclature for Mice. Genomics 1997, 45, 471-476, doi:10.1006/geno.1997.5010.
  2. Eppig, J.T. Chapter 5 - Mouse Strain and Genetic Nomenclature: An Abbreviated Guide. In The Mouse in Biomedical Research (Second Edition), Fox, J.G., Davisson, M.T., Quimby, F.W., Barthold, S.W., Newcomer, C.E., Smith, A.L., Eds.; Academic Press: Burlington, 2007; pp. 79-98.
  3. Bruford, E.A.; Braschi, B.; Denny, P.; Jones, T.E.M.; Seal, R.L.; Tweedie, S. Guidelines for human gene nomenclature. Nature genetics 2020, 52, 754-758, doi:10.1038/s41588-020-0669-3.

Reviewer 2 Report

The review by He You and Bai Lu is timely and interesting but need some corrections before publishing as follows:

1  1) In fact, the study by Maynard et al. is not the only one that investigated the contribution of Bdnf promoters to aggressive behavior. In the recent paper by Moskaliuk et al. (PMID: 36674499) was found that selective breeding for high aggression or its absence affected the Bdnf exons' transcripts in frontal cortex, midbrain and hypothalamus of Norway rats.

2   2) I think that conclusion on the implication to the anxiety-like behavior (p.10, lines 382-383) is not correct. The data presented does not support this. In the study Sakata et al. (2010) was shown that Bdnf-e4-/- mice did not exhibit anxiety-like behavior in the both elevated-plus maze (EPM) and light-dark box (LDB) tests. Recent meta-analysis by Rosso et al. (2022; PMID: 36341943) indicate that only two measures (time in open arms in EPM and time in open space in LDB) reliably detected effects of anxiolytics. So, another tests and measures for anxiety lack the construct and predictive validity. This means that anxiety-like behaviors in the OFT reported by Sakata et al. is not sufficient for judgment on any implication of BDNF promoter IV in anxiety-like behavior.

Author Response

[Please also see the attachment.]

Reviewer #2

The review by He You and Bai Lu is timely and interesting but need some corrections before publishing as follows:

Thank you for your positive feedback and valuable suggestion. We are pleased to hear that you found our manuscript timely and interesting. We appreciate your feedback and have carefully reviewed the issues you identified. We have made the necessary revisions to address your suggestions, and we hope that our updated manuscript meets your expectations.

1) In fact, the study by Maynard et al. is not the only one that investigated the contribution of Bdnf promoters to aggressive behavior. In the recent paper by Moskaliuk et al. (PMID: 36674499) was found that selective breeding for high aggression or its absence affected the Bdnf exons' transcripts in frontal cortex, midbrain and hypothalamus of Norway rats.

Thank you for your valuable feedback. We have revised our manuscript and included the recent publication by Moskaliuk et al. in section 3.5 to provide a more comprehensive overview of the research on the contribution of Bdnf promoters to aggressive behavior.

2) I think that conclusion on the implication to the anxiety-like behavior (p.10, lines 382-383) is not correct. The data presented does not support this. In the study Sakata et al. (2010) was shown that Bdnf-e4-/- mice did not exhibit anxiety-like behavior in the both elevated-plus maze (EPM) and light-dark box (LDB) tests. Recent meta-analysis by Rosso et al. (2022; PMID: 36341943) indicate that only two measures (time in open arms in EPM and time in open space in LDB) reliably detected effects of anxiolytics. So, another tests and measures for anxiety lack the construct and predictive validity. This means that anxiety-like behaviors in the OFT reported by Sakata et al. is not sufficient for judgment on any implication of BDNF promoter IV in anxiety-like behavior.

We appreciate your insightful comments. In light of your suggestion, we have revised section 3.3 of our manuscript to reflect that anxiety-like behaviors in the OFT reported by Sakata et al. may not be sufficient to draw conclusions about the implication of Bdnf promoter IV in anxiety-like behavior. We concur with your view that the construct and predictive validity of tests and measures for anxiety need to be considered carefully.

Reviewer 3 Report

The review by You and Lu describes the biology of Bdnf transcription, focusing on the different transcripts driven by its promoters. In their manuscript the authors also reported the association between the different Bdnf transcripts with some psychiatric disorders. 

The manuscript is well written and organized, reporting pertinent literature evidence. In addition to depressione, anxiety, schizophrenia, cocaine addiction and aggression, I suggest to add a paragraph reporting the importance of Bdnf also in Rett syndrome, indicating literature evidence that supports the role of this neurotrophin in the disease (such as Gao et al , 2015; Deogracias et al., 2021; Patnaik et al 2020; Sun et al, 2006).

Author Response

[Please also see the attachment.]

Reviewer #3

The review by You and Lu describes the biology of Bdnf transcription, focusing on the different transcripts driven by its promoters. In their manuscript the authors also reported the association between the different Bdnf transcripts with some psychiatric disorders. 

The manuscript is well written and organized, reporting pertinent literature evidence. In addition to depression, anxiety, schizophrenia, cocaine addiction and aggression, I suggest to add a paragraph reporting the importance of Bdnf also in Rett syndrome, indicating literature evidence that supports the role of this neurotrophin in the disease (such as Gao et al , 2015; Deogracias et al., 2021; Patnaik et al 2020; Sun et al, 2006).

Thank you for your positive feedback and valuable suggestion. We are glad that you found our manuscript well written and organized. We appreciate your suggestion to include Rett syndrome in our discussion, and we have revised our manuscript accordingly by adding a new section (section 3.5) to discuss the importance of Bdnf in Rett syndrome. We have also included the relevant literature evidence supporting the role of Bdnf in Rett syndrome, such as Gao et al. (2015), Deogracias et al. (2021), Patnaik et al. (2020), and Sun et al. (2006). Thank you again for your contribution to our manuscript.

Reviewer 4 Report

The manuscript deals with the molecular regulation and structural characteristics of the multiple Bdnf promoters, describing what is known about the role of the cellular and physiological functions of the specific Bdnf transcripts. The authors clearly analize how these promoters affect different psychiatric disorders and cognitive functions. The suggestion of future research topics to better  understand BDNF complexity is appreciable, though  updated references are needed. The number of the quoted references of the last decade is not sufficient (lees than 10 references since 2020) and the role of metal ion on BDNF expression and release could be useful to complete the picture.

The publication of the manuscript is recommended.

Author Response

[Please also see the attachment.]

Reviewer #4

The manuscript deals with the molecular regulation and structural characteristics of the multiple Bdnf promoters, describing what is known about the role of the cellular and physiological functions of the specific Bdnf transcripts. The authors clearly analyzed how these promoters affect different psychiatric disorders and cognitive functions. The suggestion of future research topics to better understand BDNF complexity is appreciable, though updated references are needed. The number of the quoted references of the last decade is not sufficient (less than 10 references since 2020) and the role of metal ion on BDNF expression and release could be useful to complete the picture. The publication of the manuscript is recommended.

Thank you for the positive feedback and the constructive suggestions. We have revised the manuscript accordingly and added a paragraph in section 2.5 to discuss the role of metal ions in BDNF expression and release. We also increased the number of references from the last decade to provide more updated information on the topic. We appreciate your recommendation for publication and hope that the revised manuscript meets your expectations.
